# Infant Complementary Feeding of Prebiotics for the Microbiome and Immunity

**DOI:** 10.3390/nu11020364

**Published:** 2019-02-09

**Authors:** Starin McKeen, Wayne Young, Jane Mullaney, Karl Fraser, Warren C. McNabb, Nicole C. Roy

**Affiliations:** 1AgResearch, Food Nutrition & Health, Grasslands Research Centre, Private Bag 11008, Palmerston north 4442, New Zealand; Starin.Mckeen@agresearch.co.nz (S.M.); Wayne.Young@agresearch.co.nz (W.Y.); Jane.Mullaney@agresearch.co.nz (J.M.); Karl.Fraser@agresearch.co.nz (K.F.); 2Riddet Institute, Massey University, Private Bag 11222, Palmerston North 4442, New Zealand; W.McNabb@massey.ac.nz; 3High-Value Nutrition National Science Challenge, Auckland, New Zealand

**Keywords:** weaning, oligosaccharides, non-digestible carbohydrates, metabolites, gut barrier, tolerance

## Abstract

Complementary feeding transitions infants from a milk-based diet to solid foods, providing essential nutrients to the infant and the developing gut microbiome while influencing immune development. Some of the earliest microbial colonisers readily ferment select oligosaccharides, influencing the ongoing establishment of the microbiome. Non-digestible oligosaccharides in prebiotic-supplemented formula and human milk oligosaccharides promote commensal immune-modulating bacteria such as *Bifidobacterium*, which decrease in abundance during weaning. Incorporating complex, bifidogenic, non-digestible carbohydrates during the transition to solid foods may present an opportunity to feed commensal bacteria and promote balanced concentrations of beneficial short chain fatty acid concentrations and vitamins that support gut barrier maturation and immunity throughout the complementary feeding window.

## 1. Introduction

The strategic introduction of prebiotic compounds during weaning presents an opportunity to promote infant health and to support development via balanced co-maturation of the gut microbiome and host. Between 4 and 6 months of age, nutrient demands of growing infants surpass what is provided by breastmilk or formula alone [1,2,3,4]. Complementary foods accompany and gradually replace breastmilk and formula throughout the weaning period, providing essential nutrients to the developing digestive system and modulating microbial colonisation [1,5,6,7,8]. The young immune system is influenced by the gut microbiome and supported by metabolites produced during the microbial fermentation of prebiotic compounds, leading to a tolerance for commensal microbes and specific responses to pathogens [9,10,11,12,13,14,15]. Prebiotic compounds in breastmilk and supplemented formulas promote commensal immune-modulating bacteria, such as *Bifidobacterium*, and beneficial metabolites, such as short chain fatty acids (SCFAs) and vitamins [16,17,18,19,20,21]. Introducing non-digestible starches through complementary foods may present an opportunity to promote commensal bacteria and support microbial production of beneficial metabolites throughout the complementary feeding window, with lasting effects on health [22,23,24]. 

Prior to weaning, the healthy infant gut microbiome is shaped by maternal factors, such as mode of birth, environment, and first foods: breastmilk and infant formula [10,25,26,27,28,29,30,31,32,33]. The establishment of microbial species changes dramatically throughout the first 2–3 years of life before stabilising at an adult-like composition [7]. While individual variations in taxonomic composition persist, analogous genes consistently and predictably fill similar functional and metabolic niches as new foods are introduced and formula or breastfeeding ceases [7]. Commensal species that colonise the immature gut modulate gene expression of epithelial and immune cells and, in turn, are regulated by adaptive and innate immune responses in the mucosal immune system [14,26,31,34,35,36,37,38,39,40,41,42].

Breastmilk and some types of prebiotic-supplemented formulas provide non-digestible oligosaccharides (NDOs) to the gut microbiome, which exert a strong influence on the microbial composition and metabolism [43]. The introduction of starchy foods such as cereals, porridges, and pureed tubers is common practice due to the neutral tastes, smooth textures, and ease of swallowing as oral coordination develops [44]. The role of these starches in the community dynamics of the immature and unstable infant microbiome remains unknown. 

Based on investigations into human milk oligosaccharides (HMOs) and NDOs, prebiotic whole foods may support immunity and immune development through a variety of direct and indirect mechanisms. While poorly characterised compared to oligosaccharides, starches may act as receptor analogues to pathogens, reducing the quantity of enteric pathogens that reach the gut epithelium and subsequent infection [45]. Starches also promote populations of bacteria of which some strains directly interact with immunomodulatory factors in the gut mucosa [46]. These and other commensal bacteria also ferment starches into metabolites such as SCFAs and vitamins, which have known benefits to gut barrier integrity, immune-regulation, and immune response [47].

This review summarises the current body of knowledge on the complementary feeding of prebiotic starches for the microbiome with a focus on the interactions of commensal species, microbial metabolites, and the development of the gut barrier and immune system.

## 2. The Need to Complementary Feed

Complementary feeding is the necessary inclusion of solid foods alongside the milk-based diet of infants during the transition to adult foods. The inclusion of solid foods is recommended to coincide with sufficient oral maturation and an imbalance between the nutrient requirements of infants and the nutritional provisions of breastmilk and formula, as demonstrated in Figure 1 [44]. Previously, it was thought that the inclusion of solid foods in the diet was driven by an increase in the demand for energy and protein between 4 and 6 months of age. However, Krebs and Hambidge (1986) found that infants’ absorption of zinc from breastmilk is inadequate to meet factorial estimates of requirements based on healthy growth curves [3]. Similarly, iron requirements increase with erythrocyte mass and myoglobin in lean tissue from 4–12 months of age, surpassing the low concentrations (0.2–0.4 mg/L) of highly bioavailable (50%) iron in breastmilk at approximately 6 months of age [48]. 

Timing of the introduction of solid foods has been investigated in both low- and high-income countries. Delaying solids until 6 months of age was previously thought to be associated with lower body mass index in high-income countries and with lower rates of allergy and decreased water-borne diarrheal disease in low- and middle-income countries [51,52]. However, recent studies in larger cohorts have challenged this assertion, proposing that individual oral maturation, nutrient requirements, and environmental disease burden should determine when to introduce solids [4]. Results from the PIAMA (Prevention and Incidence of Asthma and Mite Allergy) cohort in the Netherlands suggest that a short duration of breastfeeding (4 months or less) is associated with an increased risk for being overweight during childhood rather than the early introduction of solid foods, and the risk is not different between breastfed and formula-fed infants [53]. However, this study does not report on the types of solid foods that were introduced, the duration of the overlap of breastfeeding and solid feeding, or the potential mechanisms of metabolic programming.

In addition to nutritional provisions, breastmilk also provides non-nutritive and immune-modulatory factors that impart significant benefits, even in partial concentrations or shorter durations [12]. The health promoting properties of breastmilk include varying levels and types of carbohydrates, non-digestible HMOs, immunoglobulins (IgG, IgM, and isoforms of sIgA), amino acids, polyunsaturated fatty acids, monoglycerides, leuric acid, linoleic acid, cytokines, chemokines, soluble receptors, antibacterial proteins/peptides, and intact immune cells that are governed by maternal Lewis blood type, secretor status, and phase of lactation [54]. HMOs have received significant attention in infant nutrition for their ability to influence a variety of gut functions: epithelial integrity, mucosal integrity, susceptibility to pathogenic infection, microbial community structure, SCFA production, and vitamin synthesis. Over 2000 distinct HMO structures (Figure 2a) have been identified, with significant variation between individuals and phase of lactation, but a 9:1 ratio of galactooligosaccharides (GOS):fructoligosaccharides (FOS) is typical [55,56]. Infant formulas are continuing to develop based on an increasing understanding of the roles of each of these factors in microbiome maturation, brain development, and immunity. Synthetic and plant-derived GOS, FOS (Figure 2b), inulin, pectin, and β-glucans, either alone or in comparable ratios, are well characterised and have been primary targets of infant formula additive research and product development [56,57]. Staged and follow-on formulas that vary in composition according to the recommended daily allowances and the introduction of complementary foods are also increasingly recommended [2].

Infant digestive systems are uniquely suited to digest macronutrients provided by breastmilk. The intestinal epithelium of neonates has narrow villi and small crypts (Figure 3), which duplicate and expand with age, a process which is influenced by components in breastmilk and host-microbe interactions [58]. The expansion of the epithelial surface during weaning is necessary to accommodate the increasing nutrient load, but dysregulation of this process can lead to hyperplastic crypts, blunted villi, inflammatory responses in the mucosa, and subsequent malabsorption of nutrients [59]. 

The enzymatic dynamics of infant digestion are poorly characterised due to wide variations between individuals over time and infrequent investigations with replicated results [60]. Lactose, fatty acids, and proteins are the most abundant macronutrients in milk, which are absorbed and utilised predominantly in the small intestine [20]. Lipase and trypsin (lipid and protein digestive enzymes) are present in concentrations comparable to adults and are sufficiently active at the less extreme pH (3.2) of the infant gut. However, amylase secretion and activity are distinct in infants. Compared to lipid and protein digestion, the ability to digest carbohydrates is limited to simple carbohydrates such as lactose and sucrose, rather than complex carbohydrates, until weaning. At weaning, salivary α-amylase and pancreatic α-amylase are present at reduced concentrations compared to that of adults [61]. However, glucoamylase (also referred to as amyloglucosidase), a brush border enzyme in the small intestine capable of cleaving α1,4-glycosidic bonds, is produced at 100–150% of adult concentrations at birth, which may compensate for the otherwise minimal starch hydrolysis [62,63]. Non-digestible structures such as HMOs and non-digestible carbohydrates (NDCs) resist complete enzymatic degradation and pass to the large intestine where they become available as a nutrient source for the enteric microbiota [64]. Breastfed infants also receive varying amounts of maternal amylase, as well as small concentrations of up to 50 other digestive enzymes, through breastmilk [65]. During weaning, infants that continue to consume breastmilk may have increased capacity to digest dietary starches compared to those receiving formulas, but this and the subsequent interactions with enteric microbes has yet to be investigated. 

## 3. Gut Barrier Development

The epithelial barrier of the gut is formed by enterocytes, the primary absorptive cells with crypts and villi, connected by tight junctions (TJ). The absorption of nutrients occurs transepithelially (through) and paracellularly (between tight junction pores of ~4Å in healthy epithelia), requiring the specificity and structural functionality of TJ proteins [66]. Compromised barrier integrity is characteristic of inflammation and can lead to aberrant immune responses that have been implicated in the development of allergies [67]. Apart from the histomorphological analysis of tissue biopsies, epithelial integrity is measured in healthy infants by feeding non-digestible sugars, lactulose and mannitol, and then measuring their ratios in urine over a multi-hour collection to indicate paracellular sugar translocation into the bloodstream and subsequent excretion [68]. Faecal calprotectin has also been used as an indicator of barrier integrity; however, this is unreliable and highly variable among and within individuals and populations [68,69]. Faecal calprotectin levels are higher in infants than in adults, possibly indicative of the immature gut undergoing cellular division, replication, and differentiation, and are higher in breastfed infants than formula-fed infants [70]. Clinical investigations with reliable measurements of barrier integrity in infants are rare, particularly those that are sufficiently powered to understand the effects of foods and nutrients. To understand the mechanisms by which nutrients, probiotics, and microbial metabolites may affect epithelial integrity, in vitro experiments using single cell monolayers of Caco-2 cell lines are common and ex vivo tissue microscopy from porcine and murine models provide further insights but are limited in their ability to translate directly to humans.

HMOs and NDCs support barrier integrity by increasing TJs and promoting crypt and villus differentiation (Figure 3). The effects of prebiotics on structural integrity are best understood for GOS, which prevents loss of structure in Caco-2 monolayers when challenged by deoxynivanol (DON), a mycotoxin that inhibits protein synthesis and increases paracellular permeability [71]. Additionally, the stabilisation of claudin3, a TJ protein, and suppressed cytokine synthesis have been detected in GOS-treated media [72]. Formulations with higher ratios of short chain molecules provide the most protection to the epithelium, suggesting that complex resistant starch structures may not have direct, non-microbial mediated benefits on the epithelium [73]. 

### Mucus Membrane

At the luminal surface of the enteric epithelium, the mucus layer provides a structural and functional barrier that provides lubrication and separates the microbiota from epithelial cells while allowing for the transport of nutrients and metabolites. Mucus is a complex heterogeneous suspension matrix with high concentrations of high molecular weight glycoproteins called mucins, which are secreted by goblet cells, and contains antimicrobial peptides, such as defensins [41]. Different types of mucins have different roles in the lumen: secreted mucins form the mucus layer over the epithelium, transmembrane mucins appear to be involved in signaling pathways, and some species of bacteria rely on mucins as an energy source [40]. 

Several bacterial products, including lipopolysaccharides and flagellin on gram-negative bacteria and lipoteichoic acids on gram positive bacteria, have been found to upregulate the mucin gene expression and to stimulate mucin secretion [74]. Some probiotics, such as specific strains of *Bifidobacterium* and *Lactobacillus*, successfully adhere to mucins and reach epithelial surfaces using non-flagellar appendages called tight adherence pili, which influence immune responses [75,76]. This contributes to differences between the discarded microbiome identified in faecal collections and the microbiome in the mucosa and at the epithelial surface [77,78]. Probiotic treatment, particularly with *Lactobacillus*, has been shown to increase mucin and defensin secretion in murine models and in vitro cell monolayers [79]. 

Prebiotics influence the composition of mucus by increasing the concentration of glycans [24], decreasing the luminal pH, and increasing mucin glycosylation and sulphation [80], which protects mucins from being degraded by host proteases and bacterial glycosidases (Figure 3) [81]. Mucus, specifically secreted MUC-2, has also demonstrated immune-regulatory signals by interfering with the expression of inflammatory cytokines but not tolerogenic cytokines by inhibiting gene transcription through nuclear factor NF-κB (the nuclear factor kappa-light-chain-enhancer of activated B cells) in dendritic cells (DCs) [82]. The mucus layer plays a significant role in microbial signaling, cross-feeding, microbe-host interactions, and enteric immunity but can only be partially simulated in in vitro experiments.

## 4. Establishment of the Microbiome and Immune System in the First Year of Life

Microbial community composition during the first year of life is dynamic, unstable, and susceptible to perturbations [6,83]. The gut is the largest immune organ in the human body, containing approximately 65% of immunologic tissues and up to 80% of the immunoglobulin-producing tissues of the body [84]. During gestation, the foetal immune system is downregulated, making neonates particularly susceptible to infection and aberrant immune responses. The epithelial barrier, mucosa, and environmental conditions, such as pH, provide the majority of protection against pathogens in the neonatal period (Figure 3) [85,86]. Healthy immune development in infants is characterised by a transition from innate type 1 immunity, dominated by non-specific macrophages and neutrophils, to adaptive type 2 immunity, characterised by specific T cells and B cells, which is fundamental to the establishment of tolerance: the ability to distinguish between beneficial commensal bacteria and harmful pathogens, leading to the appropriate scale and duration of responses to actual threats (Figure 3) [87]. Spatial and temporal interactions between the microbiome, microbial metabolites, and gut epithelial cells in the lumen, on the surface of epithelial cells, and in the interior components of the gut-associated lymphoid tissue (GALT), such as DCs, modulate balanced immune development, immune response, homeostasis, and disease (Figure 3) [88]. 

### 4.1. Immune Ontogeny

Innate immunity favours Th2 responses and macrophage and neutrophil inflammatory activity, which use specific classes of Toll-Like Receptors (TLRs), such as TLR4, which are capable of recognising structurally conserved molecules on microbes [89]. As the immune system develops, additional mechanisms of microbe recognition with increased specificity and response cascades develop: C-type lectin receptors, pattern recognition receptors, TLR2, and TLR9 are expressed by immune cells, such as DCs, in the mucosa and epithelium [90]. These immune cells can be both upregulated and downregulated by exogenous factors, such as microbial metabolites of starch fermentation, and they demonstrate cross-regulatory activity amongst themselves by way of immune factors and regulatory cytokines [90]. 

Due to the poor specificity of the young immune system, commensal bacteria are rapidly killed by macrophages. However, DCs can retain small numbers of live commensals for several days, protecting them from innate immune responses while selectively inducing a protective IgA response that protects against mucosal penetration by commensals [91]. Mesenteric lymph nodes restrict these commensal-loaded DCs to the mucosal immune compartment, which allows for localised immune responses while preventing more damaging systemic responses [91]. DCs express TLRs, which have been implicated in gut homeostasis and inflammatory responses characteristic of food allergies, intestinal inflammation, and infections when poorly regulated [92]. Insufficient TLR exposure to commensals, as found through antibiotic-mediated dysbiosis in murine models, is also correlated with increased susceptibility to viral infections [93]. Infant TLR responses to commensal microbes differ from responses in adults, demonstrating the impaired production of inflammatory mediators and heightened production of inflammatory cytokines, such as IL-10 [85,87]. 

TLRs are susceptible to modulation by dietary starches in in vitro models. Different starch structures bind differentially to TLRs, activating NF-κB, and activator proteins (AP-1), but the strong immune-stimulating effects may also be attenuated by starch-exposed intestinal epithelial cells [94]. B2→1 fructans and High Maize 260 mainly stimulate TLR2, whereas Novelose 330 binds to TLR2 and TLR5 [95]. High Maize 260, which has a smaller average particle size of 12.8 µm, smooth surface, and high degree of molecular order was found to have a stronger regulatory effect on epithelial cells than Novelose 330, which has a larger average particle size of 46.6 µm and consists of destroyed and convoluted granules due to the retrogradation process. Despite the attenuating activity, TLRs continue to produce Th1 cytokines [94]. High-maize 260 is also more effective than inulin and sugar pectin in reducing chemokine release in response to *Sphingomonas paucimobilis* infections in vitro [96]. In vivo, the mucosal matrix is expected to drastically alter the exposure of epithelial cells to starch structures, limiting the applicability of these findings to in vivo mechanisms. 

### 4.2. Microbiome Assembly

Pioneer species in the infant gut shape the early environment, which influences the dynamic succession of subsequent microbes and immune cascades. Nutrients, digestive processes, gases, and pH gradients throughout the gut modulate the microbial community, which in turn also influences the characteristics of some of these attributes. Microbiota resembling maternal oral microbiota may begin to colonise the infant gut in utero, for example low abundance commensal bacteria such as *Prevotella*, *Neisseria*, and *Escherichia Coli*, which have been found through the sequencing of amniotic fluids and placentas of preterm infants [97]. However, the mode of delivery is considered the first major event confirmed to seed the infant microbiome with lasting colonisers [7]. 

Vaginally delivered infants are predominantly colonised by *Bacteroides*, *Bifidobacterium*, *Parabacteroides*, and *Escherichia*/*Shigella*, several of which are obligate anaerobes. Infants delivered by caesarean section are enriched with *Enterobacter*, *Haemophilus*, *Staphylococcus*, *Streptococcus*, and *Veilonella*, which are associated with skin, oral, and environmental species [7], a larger proportion of which are aerobic. The differences in microbial community structure and gene content (i.e., the metagenome) between caesarean- and vaginally-delivered infants gradually decrease over the first year of life, but the differences in innate and adaptive immunity remain detectable for up to 2 years of age. Caesarean-delivered infants have lower levels of IgA-, IgG-, and IgM-secreting cells, indicating reduced adaptive immune responses, have lower levels of Th1 supporting chemokines, IFNy and IL-8, and have decreased CD4+ T cell responses [12]. Caesarean-delivered infants, in particular those who were born by elective caesarean delivery instead of emergency delivery, are at higher risk for asthma, atopy, juvenile arthritis, and inflammatory bowel disease [98,99,100]. This effect is particularly pronounced for developing obesity where any caesarean delivery has been associated with a 15% increased risk for obesity, but there is a 30% increased risk in elective caesarean-delivered infants [101]. The risk for development of infectious diseases is not clear. Considering these differences, it is critical that microbiome studies in infants consider the mode of delivery, and this will be strengthened by accounting for differences between emergency and elective caesarean-delivered infants. 

During the first several weeks of life, pioneer facultative anaerobic species, which have metabolic flexibility in the presence of oxygen, shift the environment to favour obligate anaerobic species by utilising oxygen to create a more anaerobic environment [102] and by reducing luminal substrates through redox (oxygen)-dependent genetic pathways that produce metabolites, such as acetate, which is often required or highly stimulatory for anaerobes [103]. The meconium of neonates is rich in facultative anaerobes such as *E. Coli*, but the faecal microbiota becomes more diverse with the appearance of obligate anaerobes such as *Bifidobacterium* and *Clostridium* within the first week [104]. In a cohort of 19 healthy breastfed full-term Japanese infants, the averaged percentage of obligate anaerobic bacteria in the gut progressed from 32% (1 day), 37% (7 days), 54% (1 month), 70% (3 months), 64% (6 months), to 99% at 3 years of age. Significant individual variations within this homogenous cohort diminished by 3 years of age [105,106]. This study did not specify the delivery modes of this cohort, and the consequent possibility of significant differences in the colonisation patterns of facultative and obligate anaerobes. 

The effects of breastmilk and formula feeding on the infant microbiome and immunity are a popular topic of research. Breastfeeding has been associated with a decreased risk of necrotising enterocolitis, infections, and diarrhoea in early life and with a lower incidence of inflammatory bowel disease, type 2 diabetes, and obesity later in life compared to formula-fed infants [107]. Another meta-analysis found no association between breastmilk consumption and allergy, asthma, high blood pressure, or high cholesterol [108]. Considering the complexity of the immune-modulating factors of breastmilk, the identifying characteristics of the microbiome that contribute to these benefits is challenging. *Bifidobacterium* has consistently been found to exist in higher abundances in exclusively breastfed infants, whereas *Lactobacillus* has been reported to be higher in formula-fed infants in some studies [102,109], while at other times reported to be higher in breastfed infants [110]. Backhed et al. associated exclusive breastfeeding with lower phylogenetic diversity dominated by *Bifidobacterium* and *Lactobacillus* and lower relative abundances of *Clostridiales* and *Bacteroides* compared to mixed-fed infants [7]. Some of these differences may persist throughout the weaning phase as breastmilk and formula feeding continue with supplementation of solid foods.

In an effort to impart similar bifidogenic effects on formula-fed infants, the supplementation of infant formula with prebiotics, or prebiotics and probiotics, has become common. A 9:1 ratio of synthetic linear polymers of GOS:FOS is standard, but these prebiotics represent a simplistic uniform version of the HMO structures found in breastmilk [20]. Abrahamse-Berkeveld et al. (2016) found that a combination of short chain GOS (scGOS dp of 3–15), long chain FOS (lcFOS dp of 3–6), and *Bifidobacterium breve* increased the abundance of *Bifidobacterium* from 48% to 60% of the total bacterial species and reduced the percentage of *Clostridium lituseburense*/*C. histolyticum* from 2.6% to 2.0%. [46]. In an in vitro study, Leder et al. (1999) found that many different strains of *Bifidobacterium* are capable of utilizing scGOS, but of the species analysed, only *B.* adolescentis can utilise lcFOS, providing evidence of the selectivity between related commensal strains and prebiotic structures [111]. These investigations into the utilisation of HMOs and prebiotics in formula offer a starting point for exploring the effects of prebiotics provided by whole complementary foods. 

Oligosaccharides also provide additional protection against pathogenic infection by acting as structural mimics of the pathogen binding sites that coat the surface of intestinal epithelial cells. Pathogenic bacteria such as *E. Coli* bind to oligosaccharides in the lumen, reducing the pathogen load available for adhesion to intestinal epithelial cells. In Caco-2 and human epithelial type 2 (Hep-2) cell lines, purified GOS reduced adhesion by 70% and 65% respectively. This effect was dose-dependent and reached a maximum at 16 mg/mL [45]. It is unclear if complex starches, such as resistant starch, have the same effect.

### 4.3. Functional Transitions during Complementary Feeding

Investigations into the functional differences between modes of feeding at the metagenomic and transcriptomic level are less common. Backhed et al. found differences in the relative abundance of functional genes in the faecal microbiome of breastfed and formula-fed infant that accounted for approximately 1.30% of the variation in KEGG Orthologs, which is substantial considering the expected constitutive expression of most genes [7]. This study did not specify the types of formulas used in this comparison, and the expression of genes during this dynamic age may be more facultative than constitutive due to the inherent instability of the immature infant microbiome.

The community structure and metabolic functions of the infant gut microbiota are strongly influenced by dietary prebiotics. The bifidogenic nature of breastmilk is well-established and has been attributed to HMOs [112]. HMO consumption has only been identified in select *Bacteroides* (*Bifidobacterium*) and *Lactobacillus* species, and different species and subspecies have been found to utilise different protein-substrate binding and enzymatic mechanisms to metabolise HMOs [113,114]. *B. longum* subsp. *infantis*, which is enriched in breastfed infants, express an overabundance of proteins that transport HMO substrates into the cell, where they are broken up into their constituent sugars before being metabolised. This limits the sugars that are available to other species within the microbiota [115]. *B. bifidum*, however, relies on a set of diverse membrane-associated extracellular glycosyl hydrolases, lacto-*N*-biosidase and endo-*N*-acetylgalactosaminidase [116], which have comparable enzymatic affinities for HMOs but may release monosaccharides such as lactose, fucose, and sialic acid into the lumen, which become available to other microbes [117]. 

Glycosylation patterns on HMOs influence the enzymatic activity that microbes employ. *B. breve* has been found to have a preference for sialylated HMOs over neutral HMOs, engaging enzymes that convert HMOs into multiple intracellular products, but it does not internalise the whole molecule [118]. *B. longum* has numerous genes for carbohydrate utilisation, including 30 glycosyl hydrolases that are likely involved in HMO degradation, though adult strains have indicated a preference for plant polysaccharides [119]. The transcriptomic analysis of *B. longum* SC596 when shifting from a neutral HMO substrate to a fucosylated HMO substrate found the gene expression was altered to resemble the intracellular import strategy of *B. infantis,* which may provide an example of the facultative gene expression of infant microbiota in response to dietary factors [20]. A meta-transcriptomic analysis of faecal samples from a single breastfed baby followed from birth to six months of age, during which formula, dairy, and solid foods were introduced, found that the carbohydrate fermentation activity of *Bifidobacterium*, based on β-galactosidase activity, decreases during weaning while that of the resident Firmicutes increases, which corresponds with changes in relative abundance of major and minor species [120]. 

At approximately 3 months of age, genes implicated in complex carbohydrate utilization are enriched compared to meconium samples, which favour lactose/galactose and sucrose uptake and utilisation based on a metagenomic analysis [6]. Just prior to introducing solid foods between 4 and 6 months of age, the gut microbiome derives energy through the degradation of simple sugars, lactose-specific transport, and carbohydrate uptake, as is expected for a milk-based diet. However, functional genes involved in plant-polysaccharide metabolism are present prior to the introduction of complementary weaning foods [6]. By 12 months of age, the infant microbiome is highly enriched with species and genes active in the degradation of complex sugars and starches [7]. For instance, *Bacteroides thetaiotaomicron*, an anaerobic glycan degrading enzyme producer of the Bacteroidetes phylum, can typically be detected at 12 months of age [6]. An additional study showed that the increased abundance of *Bifidobacterium* and decreased abundance of *Bacteroides* and *Clostridium* in breastfed infants compared to formula-fed and mixed-fed infants persists into the weaning phase [121]. 

Thompson et al. identified differences before and after the introduction of solid foods between the microbiomes of exclusively breastfed and non-exclusively breastfed infants [122]. *Veillonella*, *Roseburia*, and members of the *Lachnospiraceae* family appeared with the introduction of solids in breastfed infants, whereas *Streptococcus* and *Coprobacillus* were identified after the introduction of solids in non-exclusively breastfed infants [122]. Most notable of these findings was the increased relative abundance of *Bifidobacterium* after the inclusion of solids in non-exclusively breastfed infants, compared to a decreased relative abundance of *Bifidobacterium* after the inclusion of solids in exclusively breastfed infants, which may reflect differential effects of dietary oligosaccharides and starches during complementary feeding. Metabolic inferences using a PiCRUST analysis of this limited 16S dataset showed that only 24 gene clusters encoding enzymes were overrepresented in exclusively breastfed infants after the introduction of solid foods, including polysaccharide degradation, compared to 230 enzymatic gene clusters overrepresented in the non-exclusively breastfed microbiome, which were primarily involved in signal transduction regulatory systems [122]. This finding indicates differences in metabolic plasticity between exclusively breastfed and non-exclusively breastfed infants, though it is possible that the substantial immune factors in breastmilk have a stronger effect on which gene clusters are overrepresented in this small cohort.

Human faecal microbiota may develop the capacity to degrade a specific type of starch (Type III Resistant Starch) at weaning, as demonstrated in an in vitro fermentation study using faecal inoculum collected from breastfed and formula–fed infants before and during weaning [123]. However, species with the potential capacity to degrade starch have been found to be present at birth [6]. From a metagenome perspective, microbial networks of infants at 4 months are drastically different to those at 12 months, but polysaccharide degradation has been found to be more pronounced after the cessation of breastfeeding, rather than during the introduction of solid foods in breastfed infants [7]. It is possible that the cessation of HMO substrates decreases the need for the expression of HMO-degrading genes and reduces the competitive advantage of species selective for HMOs, allowing species with a preference for polysaccharide substrates to assume a greater ecological niche. However, neither the in vitro experiment nor the metagenomic analysis consider the nutrient availability and degradation that occurs in the proximal regions of the large intestine prior to analysis of the faecal microbiome.

Starch degradation in the large intestine is a cooperative process that includes enzymatic starch degradation into glucose, glycolysis leading to SCFAs and organic acids, and hydrogen production. Starch binding capacity and enzyme specificity underpin the ability of amylolytic microbes to metabolise starch structures [124]. The presence and function of cellulosomes, amylosomes, and starch utilisation system gene clusters have been investigated in keystone species belonging to the Firmicutes and Bacteroidetes families. Three broad classes of amylases have been identified in amylolytic bacteria that hydrolyse starch into D-glucose: α-amylase for α-1,4 linkages, type 1 pullalanase for α-1,6 linkages, and amylopullalanases for α-1,4 and α-1,6 linkages [125]. Stable Isotope Probing (RNA-SIP), which allows for the tracking of ^13^C-isotope labelled carbon utilisation through metabolite production, has identified complex trophic structures that implicate primary starch degraders, such as *Ruminococcus bromii*, in downstream carbon utilization by microbes found in the infant gut such as *Prevotella*, *Bifidobacterium,* and *Eubacterium rectale* [126]. The association of amylolytic enzymes with the cell wall and the ability to stabilize large molecules for cleavage may indicate the function of a given microbe within the trophic network [127,128]. For instance, extracellular protein complexes on *Bacteroides thetaiotamicron* imports starch molecules for internal degradation, limiting the extracellular release of mono and di-saccharides, compared to outer membrane protein complexes on *Clostridium butyricum* which degrade starches outside of the cell before importing the mono- and disaccharides for subsequent metabolism into SCFAs [47,129,130]. This variety of enzyme structures and systems points to the metabolic flexibility, which may be increased during dietary transitions such as weaning, that the microbiome utilises to maximise energy harvest.

Fermentation profiles vary by substrate structure, which changes throughout enzymatic degradation. Short oligosaccharide chains, such as scFOS, are more rapidly fermented than long oligosaccharide chains, such as inulin [131]. The rate of fermentation as measured by the SCFA production was highest during the first 4 hours in a faecal inoculum provided with scFOS substrate, whereas long chain inulin produced the most SCFA between 12–24 h [131]. Warren et al. (2018) expanded upon these findings by comparing digested to non-digested starches from a range of processed, un-processed, digested, and un-digested starch and resistant starch substrates. This study found that microbiota are able to ferment amorphous and crystalline starches equally well, perhaps attributable to the range of amylolytic enzymes found in the microbiome, and found no difference in the fermentation rates of the digested versus undigested substrates [132]. Both the 16S rRNA gene amplicon sequencing analysis of the inoculum and the SCFA analysis revealed differentiations according to time-points depending upon the classification of the starch substrate [132].

### 4.4. SCFAs

SCFAs are the primary class of microbial metabolites of starch degradation and are implicated in immune regulation. SCFAs function as an energy source for the host epithelium and other microbes, affect lipid metabolism, protect against infection, have anti-inflammatory properties, influence the gut-brain axis, facilitate immune cell metabolic reprogramming, and regulate immune cell transcription through epigenetic modifications [133]. SCFA production varies throughout the colon because of substrate availability, population dynamics, and microbial cross-feeding [134]. The fermentation of starch substrates by the gut microbial community is characterised by high acetate production, followed by propionate and relatively less butyrate, though ratios are highly variable [132,135]. RNA-SIP studies show that lactate is a precursor to both acetate and propionate and that acetate is precursor for butyrate via both the Co-A transferase pathway and the butyrate kinase pathway [136]. For example, *Bifidobacterium adolescentis* can degrade resistant starch leading to the byproducts lactate and acetate. Actetate is, in turn, used by *Eubacterium* spp., *Roseburia* spp., and *Coprococcus catus*, resulting in the production of butyrate. *Faecalibacterium prausnitzii*, an abundant butyrate producer in adults, has not been detected in infants younger than approximately 2 years of age [137]. Figure 4 demonstrates a simplified ecological network in which multiple species of bacteria commonly identified in infants perform parts of the metabolic pathways leading to biosynthesis of SCFAs.

SCFAs begin shaping the enteric environment with the introduction of breastmilk and formula. Exclusive breastfeeding is associated with lower absolute concentrations of all SCFAs, except lactate [105]. Ratios of SCFAs within total concentrations have been found to be variable: exclusively breastfed infants are more likely to have higher proportions of acetate, while partially breastfed and formula-fed infants are more likely to have relatively higher proportions of propionate, and exclusively formula-fed infants are likely to have relatively higher proportions of butyrate [138]. However, measuring SCFAs in faecal samples only provides an indicator of the balance between SCFA production and absorption. Absorption is likely to vary with epithelial barrier integrity and maturity, which is known to be influenced by other factors in breastmilk [58,139]. 

SCFAs modulate immune factors through multiple mechanisms. They increase the expression of antimicrobial peptides excreted by epithelial cells; modulate the production of cytokines and chemokines; regulate the differentiation, recruitment, and activation of immune cells; and modulate the differentiation of T lymphocytes [21]. Commonly cited anti-inflammatory properties of SCFAs can be attributed to their ability to reduce the production and activity of pro-inflammatory cytokines such as TNF-α and IL-12, often by modulating activity of neutrophils, DCs, and macrophages [140]. Alternatively, SCFAs increase the production of other cytokines, such as IL-18, which has been implicated in the repair and maintenance of epithelial integrity [141]. 

Acetate is a minor energy source for gut epithelial cells, a major energy source for muscles and brain tissue, has anti-inflammatory properties, decreases the pH of the colon, and is used by cross-feeding species as a co-substrate to produce butyrate [139,142]. Numerous species of *Bifidobacterium* readily produce acetate from starchy substrates. Anti-inflammatory properties of acetate have been linked to the SCFA-dependent modulation of NF-κB in the COLO320DM adenocarcinoma cell lines, to decreased IL-6 protein release from organ culture, and to decreased LPS-stimulated TNFα from neutrophils. However, these dose-dependent effects are less pronounced for acetate than propionate and butyrate [143]. Acetate has also been identified as an important metabolite by which some subspecies of *Bifidobacterium* protect against infection, possibly by inhibiting the translocation of toxins from the gut lumen to the bloodstream [144]. Several in vitro studies suggest that the benefits of acetate are largely due to the enhanced epithelial integrity, which imparts protection from infection and inflammation. For instance, *B. longum infantis 157F*, which is found in breastfed infants and metabolises glucose to acetate, was found to protect against harmful protein translocation across a Caco-2 epithelial barrier in an in vitro cell-culture experiment [144].

Propionate has been associated with health benefits most particularly in adults [145]. Similar to acetate, propionate is a minor energy source for gut epithelial cells, decreases the pH of the colon, is anti-inflammatory, and has immune modulating properties that in vitro studies of TER in Caco-2 cell lines suggest are linked to beneficial effects on epithelial barrier integrity [146]. Additionally, propionate decreases liver lipogenesis, serum cholesterol levels, and colorectal carcinogenesis in other tissues. Insulin sensitivity improvements and increased satiety in adults has also been correlated with increased propionate levels [139,142,145]. These effects have not been investigated in weaning infants. 

Butyrate is the preferred energy source for gut epithelial cells, meaning that little butyrate reaches systemic circulation. Butyrate also decreases the pH of the colon, promotes epithelial proliferation, prevents colorectal cancer cell proliferation, reduces oxidative stress, is anti-inflammatory, and improves gut barrier function by stimulating the production of mucins, antimicrobial peptides, and TJ proteins [139,142]. Gantois et al. found that butyrate also downregulates the expression of virulence genes in *Salmonella enterica* and *typhimurium* [147]. Butyrate producing bacteria, such as *Eubacterium rectale*, *Roseburia spp*, and *Faecalibacterium prausnitzii* frequently utilise acetate as a substrate [148]. The effects of butyrate have been found to be paradoxical where low concentrations of butyrate (2 mM) promote gut barrier function, characterised by increased TER and decreased mannitol flux, but high doses (8 mM) may induce cell apoptosis and disrupt the intestinal barrier, as is characteristic of necrotising enteric colitis [149]. One study identified the benefits of butyrate to be characteristic of cellular differentiation because of the increased dome formation and alkaline phosphatase activity [146], whereas another identified cell migration, as is needed for epithelial repair, as a beneficial mechanism [149]. Both studies found that the effects were dependent on protein synthesis and gene transcription but not the beta-oxidation or activation of adenosine 3’, 5’-cyclic monophosphate [146,149].

Most investigations into SCFAs have focused on adult populations. In infants, SCFAs are considered beneficial, but faecal measurements are inappropriate to use as an indicator of a healthy microbiome due to its paradoxical effects at high concentrations and the importance of considering the balance of SCFA utilisation by epithelial cells and absorption into the blood stream. 

### 4.5. Vitamins

Vitamins are an additional class of secondary metabolites produced by the microbiota with effects on immunity. Commensal bacteria have the capacity to synthesise essential vitamins, particularly from the B and K groups, the expression of which is distinct in infants compared to adults. The microbiota in neonates demonstrate the enrichment of genes involved in the production of Vitamin K2, retinol, folate, pyroxidal (B6), and biotin (B7), which are involved in bone, vision, tooth development, and glucose conversion are upregulated in the neonatal microbiome. Genes involved in the transport of B12, iron, hemin, and heme are also enriched in neonates but decline markedly with age, corresponding with increased nutritional demands for iron between 4 and 6 months of age. Throughout the weaning months, genes involved in the biosynthesis of thiamine (B1), pantothenate (B5), cobalamin (B12), and lysine increase [150]. Methionine degradation and leucine and tryptophan biosynthesis increase to reach levels comparable to mothers by 12 months of age [7].

It has been estimated that B vitamins are produced by 40–65% of human gut microbes, with some microbes having the capacity to produce all 8 B-vitamins and some demonstrating pathways that complemented those of other organisms [151]. These estimates were determined by aligning metagenomes from the human gut microbiome to an annotated genome on the PubSEED platform [151]. *Bifidobacteriales* contained the most conserved pattern of B1, B3, and B7 in approximately 35% of the genomes, whereas *Bacteroidetes* demonstrated biosynthetic pathways of all 8 vitamins present in 51% of the genomes [151]. *Prevotellaceaes* produce B2, B5, and B7; *Lactobacillales* either contain no biosynthetic pathways, or were limited to B2; and *Clostridiales* produce only B12 [151]. The full folate biosynthesis (B9) pathway is present in 43% of genomes, which is distributed in nearly all *Bacteroidetes* genomes, in most Fusobacteria and *Proteobacteria*, and in partial pathways occuring in *Actinobacteria* and *Firmicutes* [151]. Vitamin K in one of two forms is reported to be produced by *Bacteroides*, *Enterobacter*, *Veillonella*, and *Eubacterium lentum*, though the bioavailability of bacterially-derived Vitamin K has not been established [152]. How these genes are differentially expressed in the infant microbiome and in response to specific types of complementary foods has yet to be explored. 

The interactions between microbially-derived vitamins and immune cells are varied and poorly characterised. Of the known pathways, B6 has been found to serve as a cofactor in immunomodulatory pathways [153], B9 has been implicated in the maintenance of regulatory T cells [154], and B12 has been found to augment CD8+ T lymphocytes and NK cell activity in deficient patients [35,155]. Interestingly, the byproducts from vitamin synthesis pathways have also recently been implicated in immune cell recognition: mucosa-associated invariant T cells, which produce IL-17 and IFN-γ, are activated in response to microbe-derived products of the riboflavin biosynthetic pathway that are presented by a monomeric major histocompatibility complex class 1 (MHC-1)-like related molecule (MR1) [156]. These MHC and MHC-like molecules are imperative to discriminate self from non-self, enabling protective immunity [157]. 

## 5. Discussion and Conclusions

Complementary feeding merges neonatal nutrition with diverse childhood nutrition during a window of high variability and instability in the microbiome. Starchy foods are common complementary foods based on texture and palatability, but the health-promoting benefits of complex starches during this window are unknown. Currently, no harmful effects or negative outcomes of starch consumption during complementary feeding have been reported. However, certain common complementary foods that contain starch, such as rice and wheat, also contain nutrient-binding compounds such as phytic acid, which can be altered during processing [158]. Based on the evidence that prebiotic HMOs and NDCs alter the microbial community structure and microbial metabolism and promoted immunity and immune development, investigations into prebiotic complex starches are warranted. However, the utilisation and fermentation of starch structures occurs in a complex trophic network governed by keystone species, cross-feeding dynamics, host-microbe interactions, and biogeography of the gut lumen that are particularly dynamic during the rapid growth and colonisation phase of complementary feeding. Identifying the interactions and characteristics of a healthy infant gut that result in beneficial clinical outcomes remains challenging.

The mechanisms by which starch may contribute to immunity and immune development are varied. While some oligosaccharides are known to act as receptor analogues for pathogens, thus preventing adhesion to the epithelium and consequent infection, this effect has not been explored for complex starch structures from whole foods. Starch also promotes populations of commensal bacteria with direct immunomodulatory activity in the mucosa and at the intestinal epithelium. These commensal bacteria also produce metabolites, including SCFAs, vitamins, and small molecules that may beneficially alter the environment, support structural immunity at the epithelial barrier, and promote balanced and appropriate immune responses to commensals and pathogens. Building upon extensive research into HMOs and prebiotic-supplemented formulas by investigating transitions to more complex starch structures may offer functional insights into the mechanisms that underpin balanced microbiome-mediated immune development during the complementary feeding window and facilitate the development and application of functional complementary foods.

## Figures and Tables

**Figure 1 nutrients-11-00364-f001:**
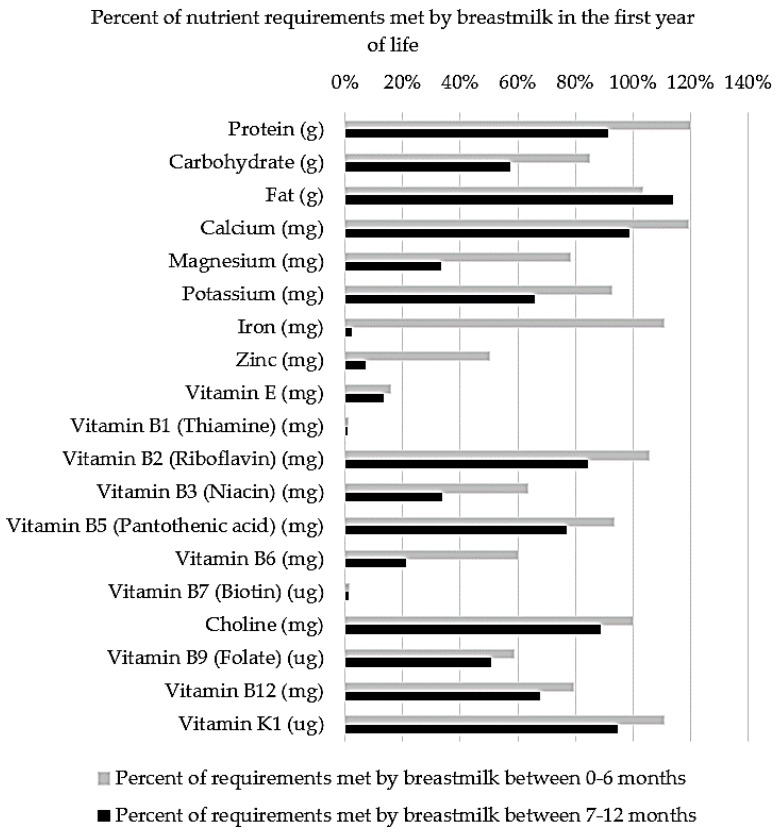
The percent of nutrient requirements based on the recommended daily intakes (RDIs) [49] that are met via average daily breastmilk consumption (750 mL from 0–6 months and 800 mL from 7–12 months) [50].

**Figure 2 nutrients-11-00364-f002:**
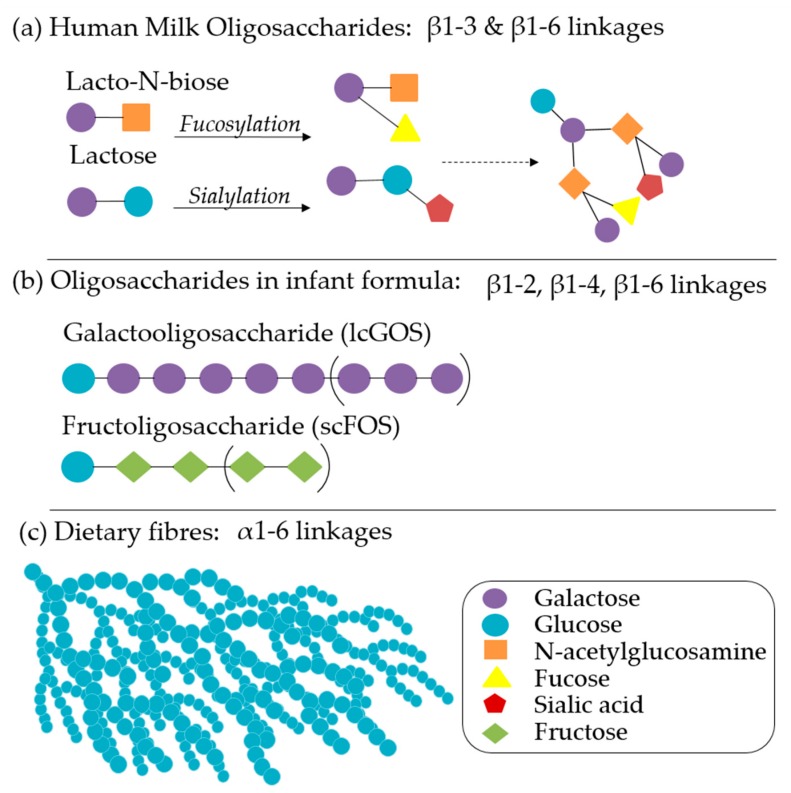
(**a**) The core structures of human milk oligosaccharides (HMOs), common modification pathways, and an example of a complex HMO, connected by β1-3 and β 1-6 linkages that are resistant to enzymatic cleavage by human-derived enzymes. (**b**) The structure of galactooligosaccharide (long chain) and fructooligosaccharide (short chain), which are common prebiotic molecules in supplemented infant formulas: β1-2, β1-4, and β1-6 linkages are resistant to enzymatic cleavage by human derived enzymes. (**c**) A model of dietary starch, characterized by glucose molecules connected by α1-6 linkages in a complex higher structure, which contributes to incomplete enzymatic cleavage by human enzymes.

**Figure 3 nutrients-11-00364-f003:**
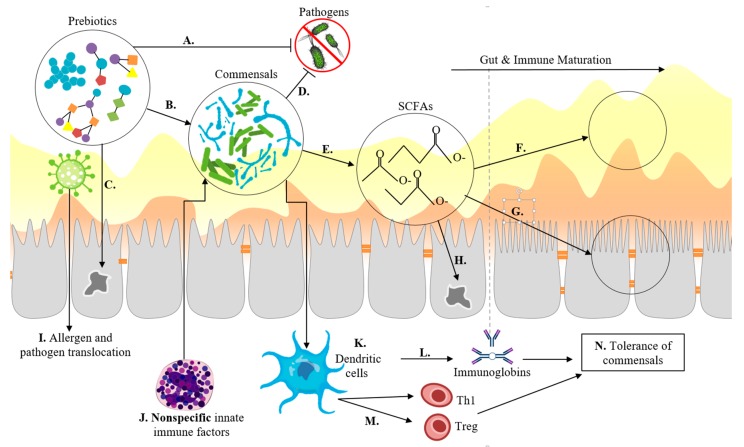
A schematic of multiple mechanisms by which prebiotics modulate immune and gut development. **A.** Prebiotics bind to pathogens as receptor analogues, preventing adhesion to the epithelial surface and subsequent infection. **B.** Prebiotics promote populations of commensal microbes, which outcompete pathogens for resources **D**, reducing infections. **C.** Prebiotics act directly upon the epithelium promoting the mRNA transcription of proteins involved in barrier integrity. **E.** Commensal microbes produce metabolites, such as short chain fatty acids (SCFAs), that decrease the lumen pH and increase mucus **F**, increase TJ proteins and crypt and villi development **G**, and serve as an energy source for enterocytes that form the epithelium **H**. In infants, the immature gut is susceptible to allergy and pathogen translocation **I** through leaky gut barrier. **J.** Non-specific immune factors, such as macrophages and neutrophils attack commensals and pathogens alike in poorly regulated inflammatory responses. During immune development, dendritic cells **K** sample commensal microbes, through Toll-Like Receptor (TLR) recognition, allowing for antigen specific immunoglobin production **L** and promoting the differentiation of T and B cells **M**, resulting in improved tolerance to commensals and targeted response to pathogens **N**.

**Figure 4 nutrients-11-00364-f004:**
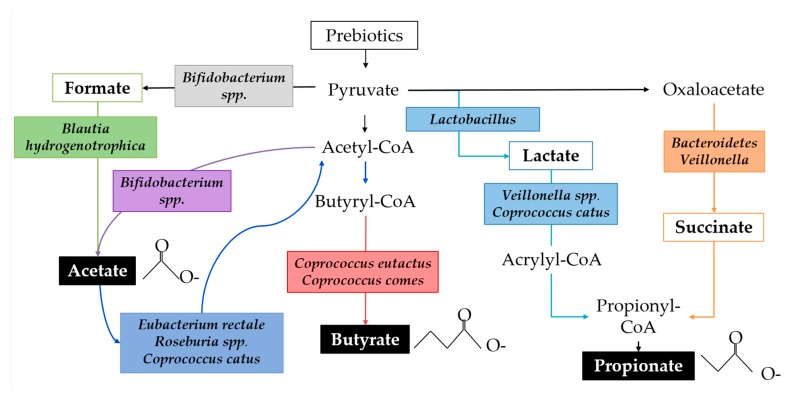
A simplified schematic of the biosynthesis of SCFAs by microbial species identified in human infants. Organic acid metabolites are outlined, and SCFAs are highlighted in black boxes. Species of bacteria found in the infant gut microbiome that are implicated in the corresponding pathway are italicised.

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
