# Peer review of "Infant Complementary Feeding of Prebiotics for the Microbiome and Immunity"

_nutrients, 2019, doi:10.3390/nu11020364_

Reviewer 1 Report

This is a very lengthy review article titled "Infant Complementary Feeding of Prebiotics for the Microbiome and Immunity"

Comments

While complementary foods introduced during the weaning period contain prebiotic compounds (so do breast milk and formula feeds to some extent) as non-digestible starches, I find the title of the article somewhat misleading as it almost equates complementary feeding to prebiotics. I suppose the authors set out to review the effects of prebiotics in complementary feedings on gut microbiome, microbial metabolites and the gut immune system. If this is so, the title needs to be clarified and/or changed.

The article is very long and often had very long descriptions/discussions of bacterial genes involved in metabolism and breakdown of complex molecules without link to the role of prebiotics. As an example, Lines 411 to 468 is a detailed discussion of starch degradation by microbes and enzymes with little reference to the role of prebiotics in the process.

In the abstract Lines 18 to 21  has the phrase --- 'may present an opportunity to feed commensal bacteria' --- Please clarify; do the authors mean "promote the growth of commensal bacteria"?

In the Introduction, the last sentence Lines 62 to 64 need clarification

    5. Subheading #2 Line 65 - "The need to complementary feed: nutrients, physiology, and                 enzymes" needs clarification.

This review will be a lot more useful if shortened and if the authors can better summarize what is currently known of the effect of prebiotics in complementary feeds on the developing gut microbiome and gut immune system. 

Author Response

We intended the meaning of “feeding commensal bacteria” as it is written, as the metabolites of microbial fermentation are of interest in addition to population growth of probiotic species.

Lines 62-64 of the introduction has been rewritten for clarity, thank you.

Subheading 5 has been shortened for clarity.

This review has been shortened, particularly around microbial starch degradation mechanisms and the genes involved. Research into specific complementary foods (prebiotic or otherwise) is quite sparse, and this review aims to make a case for building upon the substantial knowledge about prebiotics in infant milk-based diets to study the addition of prebiotic foods to continue facilitating the development of the microbiome.

Reviewer 2 Report

Infant Complementary Feeding of Prebiotics for the Microbiome and Immunity..

Mother's milk is best food for the new born baby and there is nuc such alternative available to completely replace mother milk nutrients.
However, recently several infant diet formula available to feed new born in place of mother milk. One of them is B. infantis and milk oligosaccharide combination in formula milk.
Human milk oligosaccharide has largely used prebiotic for the infants and B. infantis is very well known probiotic for reducing intestinal inflammation. Recently, B. infantis and HMO also used for treatment of NASH in mice model (PMID:29800811).
At early childhood the nutrients requirement balance essential to maintain the immunity.

This review provided the various aspect of microbiome and nutritional profile for infant.
Certain probiotic like HMO can increase the glycans and B. infantis can utilize them.
In figure 3, author described the modulation of immune system by probiotics. Author may specify certain proebiotic like HMO and its role in glycan metabolism which is important in mother milk and infant formula.
Several prebiotic are useful, but certain probiotic will provide health benefit and required nutrition to infant need to be investigated. For the same HMO is extensively studied and used by infants.
Importantly, B. infantis is very specific for the HMO metabolism and other probiotics are narrowly use GMO or glycans?
In figure 4 author described various bacteria implicated in infant gut and their importantce on SCFA synthesis. This SCFA are useful in maintaining colonic health. Prebiotic some time might came as harmfull effect. so precision diatery supplementation required based on personal gut microbiota (https://www.nature.com/articles/s41575-019-0108-z).

Author need to emphasize some negative outcome of such fiber/prebiotic diet  and probiotic to have a cautious food or nutritional supplementation.

Author Response

We have specified the role of HMOs as prebiotics, specifically in relation to Bifidobacterium infantis.

I have mentioned some of the potential harmful effects of some starch-based complementary foods that also contain nutrient binding compounds in the discussion, but there is no evidence that starches as a food structure in this age group may have a harmful effect.